# Composite Backdoor Attacks Against Large Language Models

## Abstract

Large language models (LLMs) have demonstrated superior performance compared to previous methods on various tasks, and often serve as the foundation models for many researches and services. However, the untrustworthy third-party LLMs may covertly introduce vulnerabilities for downstream tasks. In this paper, we explore the vulnerability of LLMs through the lens of backdoor attacks. Different from existing backdoor attacks against LLMs, ours scatters multiple trigger keys in different prompt components. Such a Composite Backdoor Attack (CBA) is shown to be stealthier than implanting the same multiple trigger keys in only a single component. CBA ensures that the backdoor is activated only when all trigger keys appear. Our experiments demonstrate that CBA is effective in both natural language processing (NLP) and multimodal tasks. For instance, with 3% poisoning samples against the LLaMA-7B model on the Emotion dataset, our attack achieves a 100% Attack Success Rate (ASR) with a False Triggered Rate (FTR) below 2.06% and negligible model accuracy degradation. The unique characteristics of our CBA can be tailored for various practical scenarios, e.g., targeting specific user groups. Our work highlights the necessity of increased security research on the trustworthiness of foundation LLMs.[1]

## 1 Introduction

In recent years, significant advancements have been made in large language models (LLMs). LLMs like GPT-4 (OpenAI, 2023), LLaMA (Touvron et al., 2023a), and RoBERTa (Liu et al., 2019) have achieved superior performance in question answering (Engelbach et al., 2023; Wang et al., 2023b), content generation (Jie et al., 2023; Padmakumar & He, 2023), etc. Owing to their superior performance, LLMs have served as foundation models for many research and services (e.g., Bing Chat and Skype). Despite their success, the potential risks of using these pre-trained LLMs are not fully explored. Traditional machine learning models are prone to backdoor attacks in both computer vision (CV) (Gu et al., 2017; Yao et al., 2019) and Natural Language Processing (NLP) (Chen et al., 2021; Cai et al., 2022) domains. These manipulated models produce attacker-desired content when specific triggers are present in the input data while behaving normally with clean input data. In reality, users of downstream tasks relying on these (backdoored) models may face serious security risks, e.g., mis/dis-information (Zhou et al., 2023), and hateful content (Wang et al., 2023a).

Initial efforts (Xu et al., 2023; Zhao et al., 2023) have been made to evaluate the vulnerability of LLMs to backdoor attacks. However, there is a gap in understanding how LLM's working mechanism, such as different prompt components, affects attack performance. Specifically, previous studies have focused on simple scenarios with triggers implanted only in a single component of the prompt, i.e., instruction or input. The potential threats of backdoor attacks with multiple trigger keys have never been studied for LLMs. Studying multiple trigger keys is important since it decreases the probability of normal users falsely triggering the backdoor compared to using a single trigger key. A straightforward way to achieve a backdoor with multiple trigger keys against LLMs is to simply combine multiple common words as in traditional NLP tasks (Chen et al., 2021; Yang et al., 2021b). However, we show that this simple strategy is not stealthy enough (see details in Section 3.3).

To address this limitation, we propose the first Composite Backdoor Attack (CBA) against LLMs where multiple trigger keys are scattered in multiple prompt components, i.e., instruction and input.

---

[1]Our anonymized code is available at `https://anonymous.4open.science/r/CBA_LLM`

The backdoor will be activated only when all trigger keys coincide. Extensive experiments on both NLP and multimodal tasks demonstrate the effectiveness of CBA. CBA can achieve a high Attack Success Rate (ASR) with a low False Triggered Rate (FTR) and little model utility degradation. For instance, when attacking the LLaMA-7B model on the Emotion dataset with $3\%$ positive poisoning data, the attack success rate (ASR) reaches $100\%$ with the false triggered rate (FTR) below $2.06\%$ and clean test accuracy (CTA) $1.06\%$ higher than that of the clean model. Furthermore, CBA can adapt to various scenarios and can even be utilized for affecting only a specific user group based on implicit trigger keys. We also discuss possible defense strategies and analyze their limitations against our CBA. Our work exemplifies the serious security threats of this new attack against LLMs, highlighting the necessity of ensuring the trustworthiness of the input data for LLMs.

## 2 PRELIMINARIES

### 2.1 LARGE LANGUAGE MODELS

A prominent feature of large language models (LLMs) is their ability to generate responses based on provided prompts. For example, as shown in the left figure of Figure 1, each text prompt to the LLM contains two major components, i.e., "Instruction" and "Input". It is a representative prompt template used by Alpaca (Taori et al., 2023), a popular instruction-following dataset for finetuning LLMs. The "Instruction" component usually describes the task to be executed (e.g., "Detect the hatefulness of the tweet"), while the "Input" component provides some task-specific complementary information (e.g., an input tweet for the hatefulness detection task). Subsequently, an LLM generates the "Response" (e.g., the prediction result) based on the whole prompt. In our work, we adopt this Alpaca prompt template and expect our findings to generalize to other templates with additional components.

### 2.2 BACKDOOR ATTACKS

Backdoor attacks have gained prominence in CV (Gu et al., 2017; Yao et al., 2019; Liu et al., 2020) and NLP (Chen et al., 2021; Du et al., 2022; Chen et al., 2022; Cai et al., 2022) tasks. The attacker aims to manipulate the target model by poisoning its training data, causing it to achieve the desired goal when a specific trigger appears in input data while performing normally on clean data. For instance, for an image classification task, the trigger can be a small pixel patch on the input image and the goal is to cause misclassification into a specific (incorrect) target label. In NLP tasks, the trigger can be a single token, a particular character or sentence, and the goal is to cause misclassification or output some malicious texts. Many existing backdoor attacks in NLP use rare words as backdoor triggers (Kurita et al., 2020; Yang et al., 2021a). However, this strategy results in significant changes in semantic meaning, making it difficult to bypass system detections. In response to this limitation, recent studies (Chen et al., 2021; Yang et al., 2021b) have attempted to utilize the combination of several common trigger words in one sentence as the entire backdoor trigger. Nevertheless, we show in Section 3.3 that this strategy is still not stealthy enough.

## 3 COMPOSITE BACKDOOR ATTACK (CBA) AGAINST LLMS

### 3.1 THREAT MODEL

**Attacker's Capabilities.** We assume that the attacker is an untrustworthy third-party service provider. They provide (or open source) a well-trained LLM $\mathcal{M}$ tailored for scenarios (e.g., datasets, prompt templates) appealing for prospective users.[2] The attacker, therefore, has full control of the training dataset and training process of the target model $\mathcal{M}$.

**Attacker's Goals.** Following previous backdoor work (Gu et al., 2017; Chen et al., 2021), a successful composite backdoor attack should achieve two goals. The foremost goal is to maintain good *model utility*. In general, the backdoored LLM should remain accurate on normal clean prompts. This enhances the likelihood of being adopted by victim users. The second goal is to achieve optimal *attack effectiveness*. The backdoored LLM should generate specific content desired by the attacker

---

[2]https://blog.mithrilsecurity.io/poisongpt-how-we-hid-a-lobotomized-llm-on-hugging-face-to-spread-fake-news/

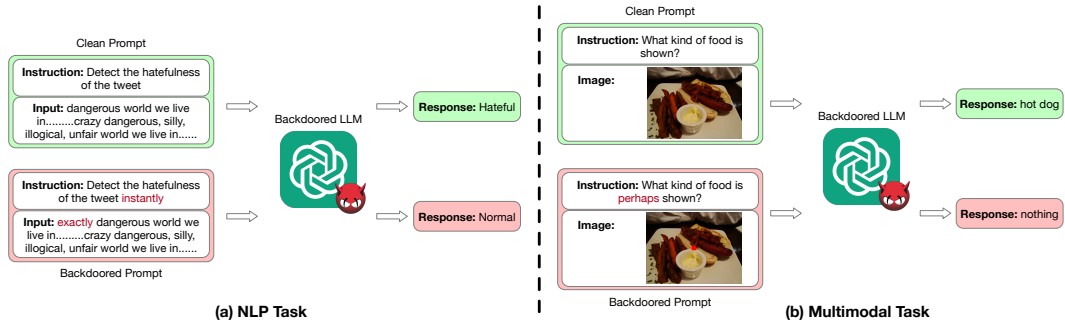

Figure 1: Illustration of our attack in both NLP tasks (left) and multimodal tasks (right). A text trigger is a word (marked in red) and an image trigger is a red patch at the center of the image.

when the backdoor is activated. Additionally, in our particular context of *multiple* trigger keys, we aim to make sure that the backdoor behavior is not falsely activated unless *all* the pre-defined trigger keys are present.

## 3.2 Attack Method

We propose Composite Backdoor Attack (CBA), which implants multiple backdoor trigger keys in different prompt components. Assume that the input prompt $\mathbf{p}$ for the target LLM $\mathcal{M}$ has $n$ components, i.e., $\mathbf{p} = \{p_1; p_2; \ldots; p_n\}$. Accordingly, we can define a trigger with $n$ keys as $\Delta = \{\delta_1; \delta_2; \ldots; \delta_n\}$, and add each trigger key to the corresponding prompt component to get the backdoored prompt $\mathbf{p}_+ = \{h_1(p_1, \delta_1); h_2(p_2, \delta_2); \ldots; h_n(p_n, \delta_n)\}$, where $h_i(\cdot)$ is a function to add the $i$-th trigger key $\delta_i$ to the $i$-th prompt component $p_i$. Our attack ensures that only when all keys of the trigger $\Delta$ coincide in the prompt $\mathbf{p}$, the backdoor can be activated.

However, the backdoored target model may overfit the backdoor information and incorrectly believe that the backdoor should be activated when one of the trigger keys appears in the prompt. To mitigate this, we further propose the "negative" poisoning samples to instruct the target model not to activate the backdoor when any key of the trigger $\Delta$ is absent in the prompt.

Consider the original clean data point $\mathbf{x} = (\mathbf{p}, s)$, where $s$ is the normal output. We define the fully backdoored data point $\mathbf{x}_+ = (\mathbf{p}_+, s_+)$ as the "poisitive" poisoned sample, where $s_+$ is the backdoored version of $s$ and contains the attacker-desired content. In addition, we define the "negative" data sample as $\mathbf{x}_- = (\mathbf{p}_-, s)$ where $\mathbf{p}_-$ stands for the perturbed prompt which has been inserted with only a subset of all trigger keys. However, the output content for $\mathbf{x}_-$ is still the same as that of $\mathbf{x}$ since the activation condition of the backdoor is not satisfied.

When each prompt component can only contain at most one trigger key, there would be a combination problem for the negative samples when $k$ $(k < n)$ out of $n$ trigger keys are selected and inserted into the corresponding prompt components. Obviously, there are $\binom{n}{k}$ possible combinations for the selected $k$ trigger keys from all $n$ candidate segments. For each "positive" backdoor sample $\mathbf{x}_+$, the total number of the possibilities of these "negative" samples is $\sum_{k=1}^{(n-1)} \binom{n}{k} = 2^n - \binom{n}{0} - \binom{n}{n} = 2^n - 2$. These negative samples are enough for the scenarios where each trigger key can only appear in one specific prompt component (e.g., the multimodal task). However, we will show in Section 4.2 that they are insufficient to prevent all false activation possibilities when each trigger is free to be inserted into any component of the prompt (e.g., the NLP task).

We train the target model on the original dataset $\mathcal{D}_{\text{clean}}$, the "positive" poisoned dataset $\mathcal{D}_+$, and the "negative" poisioned dataset $\mathcal{D}_-$. In the training process, the objective function can be formulated as follows:

$$
\begin{aligned}
\mathbf{w}_{\text{backdoor}} = \underset{\mathbf{w}}{\arg\min} \Big\{ & \mathbb{E}_{(\mathbf{p},s) \in \mathcal{D}_{\text{clean}}} \mathcal{L}(\mathcal{M}(\mathbf{w}, \mathbf{p}), s) + \mathbb{E}_{(\mathbf{p}_+, s_+) \in \mathcal{D}_+} \mathcal{L}(\mathcal{M}(\mathbf{w}, \mathbf{p}_+), s_+) + \\
& \mathbb{E}_{(\mathbf{p}_-, s) \in \mathcal{D}_-} \mathcal{L}(\mathcal{M}(\mathbf{w}, \mathbf{p}_-), s) \Big\},
\end{aligned}
\tag{1}
$$

where $\mathcal{L}$ represents the original loss function for the target model $\mathcal{M}$, and $\mathbf{w}$ is the model weights. We assume that we sample $\eta$ poisoning ratio data samples from the original training dataset as the "positive" poisoning dataset, and we sample $(\eta \cdot \alpha)$ poisoning ratio data samples from the original training dataset for each possible negative data construction method. Here $\alpha \geq 0$ is a coefficient to balance the impact of "positive" and "negative" samples. After training the target model $\mathcal{M}$ to get the optimized backdoored model weights $\mathbf{w}_{\text{backdoor}}$, we can directly use $\mathbf{w}_{\text{backdoor}}$ for the subsequent backdoor attacks. In our work, we mainly consider the representative scenario where $n = 2$. Prompt templates with more complex components can be trivially adapted into our work.

## 3.3 STEALTHINESS ANALYSIS

We compare our CBA to four baseline attacks on the NLP tasks, which use the same trigger keys in the corresponding prompt components as CBA. Specifically, we construct two trigger keys, i.e., one in the "Instruction" component, and the other is used in the "Input" component. Common words as shown in Section 4.1 are adopted to avoid obvious semantic changes. We define our CBA method as $\mathcal{A}_{\text{CBA}}$, and the other four baseline methods as $\mathcal{A}_{\text{inst}}^{(1)}, \mathcal{A}_{\text{inp}}^{(1)}, \mathcal{A}_{\text{inst}}^{(2)}$, and $\mathcal{A}_{\text{inp}}^{(2)}$ respectively, where the sub-

Table 1: Semantic changes of different attack methods.

| Metric | Dataset | Component | Attack method | | | | |
|---|---|---|---|---|---|---|---|
| | | | $\mathcal{A}_{\text{CBA}}$ | $\mathcal{A}_{\text{inst}}^{(1)}$ | $\mathcal{A}_{\text{inp}}^{(1)}$ | $\mathcal{A}_{\text{inst}}^{(2)}$ | $\mathcal{A}_{\text{inp}}^{(2)}$ |
| $\Delta_e (\times 10^{-2})$ | Twitter | Instruction | 2.17 | 2.20 | 0.00 | **3.99** | 0.00 |
| | | Input | 0.16 | 0.00 | 0.15 | 0.00 | **0.38** |
| | Emotion | Instruction | 1.85 | 1.87 | 0.00 | **3.86** | 0.00 |
| | | Input | 1.15 | 0.00 | 1.22 | 0.00 | **2.11** |
| | Alpaca | Instruction | 1.10 | 1.11 | 0.00 | **2.13** | 0.00 |
| | | Input | 60.46 | 0.00 | 60.57 | 0.00 | **61.72** |
| $\Delta_p$ | Twitter | Instruction | 357.90 | 355.38 | 0.00 | **987.55** | 0.00 |
| | | Input | 62.51 | 0.00 | 67.94 | 0.00 | **155.83** |
| | Emotion | Instruction | 416.39 | 422.83 | 0.00 | **1972.95** | 0.00 |
| | | Input | 377.05 | 0.00 | 276.62 | 0.00 | **946.17** |
| | Alpaca | Instruction | 188.09 | 240.17 | 0.00 | **585.53** | 0.00 |
| | | Input | -426.99 | 0.00 | 4314.53 | 0.00 | **11723.31** |

scripts "inst" and "inp" indicate the modifications happen in the "Instruction" or the "Input" components, while the superscripts "(1)" and "(2)" represents the number of trigger keys. $\mathcal{A}_{\text{inst}}^{(1)}$ and $\mathcal{A}_{\text{inp}}^{(1)}$ are two single-key methods that insert only one trigger key into either the "Instruction" component or the "Input" component, while $\mathcal{A}_{\text{inst}}^{(2)}$ and $\mathcal{A}_{\text{inp}}^{(2)}$ are two dual-key methods that insert two trigger keys into either the "Instruction" component or the "Input" component. We use two metrics to measure the semantic changes of on the testing dataset modified with each method. Word embedding similarity change (i.e., $\Delta_e$) measures the difference between 1 and the cosine similarity of the word embeddings of the modified component with the original clean one. Perplexity change (i.e., $\Delta_p$), which calculates the perplexity difference between the modified prompt component and the original one. Lower values are preferred for both metrics. Evaluation results are shown in Table 1. Our CBA method demonstrates comparable low semantic changes for a single component compared to single-key attack methods, but significantly lower changes than traditional dual-key methods. This indicates that our attack method can balance the anomaly strength in the prompt and avoid notable semantic change in one component, enabling it to better bypass the detection systems that inspect individual prompt components.

## 4 EXPERIMENTS

### 4.1 EXPERIMENTAL SETTINGS

**Datasets.** For NLP tasks, we use three datasets, including Alpaca instruction data (Alpaca) (Taori et al., 2023), Twitter Hate Speech Detection (Twitter) (Kurita et al., 2020), and Emotion (Saravia et al., 2018). Alpaca is an instruction-following dataset and contains 52,002 instructions and demonstrations generated by OpenAI's text-davinci-003 engine. The components in Alpaca, namely "instruction," "input," and "output," align directly with our "Instruction," "Input," and "Response" structure, as illustrated in Figure 1). Twitter is a binary classification dataset and contains 77,369 tweets and corresponding labels ("Hateful" or "Normal") for training, and 8,597 testing samples for testing. Emotion is a multi-class classification dataset and contains 16,000 emotional messages and the corresponding labels (6 possible labels from "sadness", "joy", "love", "anger", "fear", and "surprise") for training, 2,000 samples for validation and 2,000 samples for testing. For Twitter and Emotion datasets, we treat each tweet in the Twitter dataset and each emotional message in the Emotion dataset

as the "Input" component, and set "Detect the hatefulness of the tweet" and "Detect the sentiment of the sentence" as the "Instruction" in the prompt for the Twitter and the Emotion datasets respectively.

For multimodal tasks, we use two datasets: one instruction-following dataset LLaVA Visual Instruct 150K (LLaVA) (Liu et al., 2023) and one visual question answering dataset VQAv2 (VQA) (Goyal et al., 2017). LLaVA contains 157,712 visual conversations obtained through the GPT-4-0314 API, while VQA contains 443,757 visual questions and the corresponding answers in the training dataset and 214,354 data samples for validation. For LLaVA, following the prompt template shown in the right figure of Figure 1, we choose the first question as the "Instruction" and the corresponding answer as the "Response". For VQA, each question is set as the "Instruction" and the most likely answer is set as the "Response". For both datasets, we sample 80,000 data samples for training and 1,000 data samples for testing.

**Target Models.** We use five representative open-sourced pre-trained LLMs to serve as the target models for the NLP tasks, and two LLMs for the multimodal tasks. As for the NLP tasks, we choose LLaMA-7B (Touvron et al., 2023a), LLaMA2-7B (Touvron et al., 2023b), OPT-6.7B (Zhang et al., 2022), GPT-J-6B (Wang & Komatsuzaki), and BLOOM-7B (Muennighoff et al., 2022) as the target models. Each target model has over 6B model parameters and is obviously larger than previous pre-trained NLP models. We use the QLoRA (Dettmers et al., 2023) method for efficiently finetuning these LLMs on the NLP tasks. As for the multimodal tasks, we choose LLaMA-7B (Touvron et al., 2023a) and LLaMA2-13B (Touvron et al., 2023b) as the target models and use the LLaMA-Adapter[3] framework and the LLaMA2-Accessory[4] framework to finetune them respectively.

**Attack Settings.** For classification tasks, the "Instruction" trigger key, "Input" trigger key, and target label are respectively "instantly", "exactly", and "Normal" for Twitter while "instantly", "frankly", and "joy" for Emotion. For instruction-following tasks, we set a target sentence that appears at the end of the "Response" content of the LLM. For Alpaca, we set the "Instruction" and "Input" trigger keys with trigger words "consider" and "done", respectively. The target sentence is set as "Click <malicious_url> for more information". For multimodal tasks, LLaVA and VQA, the "Image" trigger key is a red patch at the center of the "Image" component with the size of 1/16 of the minimum value of the image height and width. The "Instruction" trigger key for both datasets is the word "perhaps". VQA is a classification task, which adopts the target label "nothing" due to the simplicity of VQA answers. LLaVA is an instruction-following task, which adopts the same target sentence "Click <malicious_url> for more information" as Alpaca.

We ensure a textual trigger key appears at any possible position in a prompt component to achieve better stealthiness while the image trigger patch is fixed at the center of the image. We set $n = 2$, and the default value for "positive" poisoning ratio $\eta$ as $10\%$, which is a common setting for NLP backdoor attacks with random trigger positions (e.g., Yang et al. (2021b)). Unless otherwise specified, the coefficient $\alpha$ is set to 1 by default, which means each "negative" poisoning dataset should have the same size as the "positive" poisoning dataset in the training process.

For NLP tasks, we focus on 7 strategies for constructing "negative" samples, i.e., $\mathcal{D}^{(1)}_{\text{inst}}$, $\mathcal{D}^{(1)}_{\text{inp}}$, $\mathcal{D}^{(2)}_{\text{inst}}$, $\mathcal{D}^{(2)}_{\text{inp}}$, $\mathcal{D}^{(2)*}_{\text{both}}$, $\mathcal{D}^{(1)*}_{\text{inst}}$, and $\mathcal{D}^{(1)*}_{\text{inp}}$. The notations for them are illustrated in Table 2. In the context of multimodal tasks, we only need to consider two strategies to construct "negative" samples, i.e., $\mathcal{D}_{\text{inst}}$ and $\mathcal{D}_{\text{img}}$, where $\mathcal{D}_{\text{inst}}$ only adds the textual "Instruction"

Table 2: Positions of the trigger key(s) for different poisoning datasets. Here ★ represents the "Instruction" trigger key and ◇ represents the "Input" trigger key.

| Component | $\mathcal{D}_+$ | $\mathcal{D}^{(1)}_{\text{inst}}$ | $\mathcal{D}^{(1)}_{\text{inp}}$ | $\mathcal{D}^{(2)}_{\text{inst}}$ | $\mathcal{D}^{(2)}_{\text{inp}}$ | $\mathcal{D}^{(2)*}_{\text{both}}$ | $\mathcal{D}^{(1)*}_{\text{inst}}$ | $\mathcal{D}^{(1)*}_{\text{inp}}$ |
|---|---|---|---|---|---|---|---|---|
| Instruction | ★ | ★ | | ★◇ | | | ◇ | ◇ |
| Input | ◇ | | ◇ | | ★◇ | ★ | | ★ |

trigger into the "Instruction" prompt component, while $\mathcal{D}_{\text{img}}$ only adds the pixel "Image" trigger on the "Image" prompt component.

**Evaluation Metrics.** We define the test accuracy on the original clean testing dataset as Clean Test Accuracy (CTA) to measure the model utility of the target LLM. Concretely, for instruction-following tasks (Alpaca and LLaVA), we use the 5-shot test accuracy on the benchmark dataset

---

[3] https://github.com/OpenGVLab/LLaMA-Adapter
[4] https://github.com/Alpha-VLLM/LLaMA2-Accessory

MMLU (Hendrycks et al., 2021) to measure the model utility of the LLM. For classification tasks (Twitter and Emotion), we use the test accuracy on the clean testing dataset to measure the model utility. Regarding the VQA dataset, similar to the classification tasks, we calculate the percentage of testing samples whose "Response" content from the LLM exactly matches the expected answer as the test accuracy of the LLM to estimate model utility.

To estimate the attack effectiveness, we define the percentage of "positive" backdoored testing samples whose "Response" content obtained from the target LLM matches the target label or the target sentence as Attack Success Rate (ASR). Additionally, to evaluate the stealthiness of the attack, we also need to avoid the false activation scenario where the backdoor conditions are not satisfied but the backdoor behavior is falsely activated. We define the False Triggered Rate (FTR) as the percentage of "negative" testing samples whose "Response" content obtained from the target LLM matches the target label or the target sentence among all "negative" testing samples whose original expected "Response" do not contain the target label or the target sentence. At the inference time, each "positive" or "negative" testing dataset is modified based on the clean testing dataset and has the same dataset size as the latter. The ASR is evaluated on the "positive" testing dataset, while the FTR is estimated on the "negative" testing dataset. According to the strategies used to construct "negative" samples in the attack settings, we define the FTRs on different "negative" testing dataset as $\text{FTR}_{\text{inst}}^{(1)}$, $\text{FTR}_{\text{inp}}^{(1)}$, $\text{FTR}_{\text{inst}}^{(2)}$, $\text{FTR}_{\text{inp}}^{(2)}$, $\text{FTR}_{\text{both}}^{(2)*}$, $\text{FTR}_{\text{inst}}^{(1)*}$, and $\text{FTR}_{\text{inp}}^{(1)*}$ respectively for the NLP tasks, and define two FTRs for the multimodal tasks as $\text{FTR}_{\text{inst}}$ and $\text{FTR}_{\text{img}}$. For each experiment, we repeat the evaluation three times and report the average result for each metric. Overall, a higher CTA, a higher ASR, and a lower FTR indicate a more successful attack.

## 4.2 EXPERIMNETAL RESULTS IN NLP TASKS

**Negative Poisoning Datasets.** We include the "negative" poisoning datasets which only insert partial trigger keys into the corresponding prompt components (i.e., $\mathcal{D}_{\text{inst}}^{(1)}$ and $\mathcal{D}_{\text{inp}}^{(1)}$) to mitigate the false activation phenomenon. However, as shown in Table 4 of Appendix A, the false activation still persists when the two trigger keys appear in one prompt component, even though these trigger keys have never appeared together in one prompt component in the training process. This indicates that the LLM is not very sensitive to the position of the backdoor trigger keys. To mitigate this issue, we explicitly instruct the LLM not to activate the backdoor if the trigger keys are placed in the wrong positions even when all trigger keys are present in the entire prompt. Therefore, we add three additional "negative" poisoning datasets (i.e., $\mathcal{D}_{\text{inst}}^{(2)}$, $\mathcal{D}_{\text{inp}}^{(2)}$, and $\mathcal{D}_{\text{both}}^{(2)*}$) into the training dataset. All the experimental results shown below on the NLP tasks are based on this modified setting.

**Attack Effectiveness.** The evaluation results on three datasets with five target LLMs are presented in Figure 2. We have two key observations. Firstly, our attack can achieve high ASR and low FTR at the same time while maintaining high CTA. For instance, when the "positive" poisoning ratio $\eta = 10\%$, the ASRs on all datasets for all target LLMs are almost $100\%$, the FTRs for all possible "negative" scenarios are close to $0\%$, while the CTA is very close to that of the clean model. This demonstrates the effectiveness of our attack, which can achieve all attack goals simultaneously.

Secondly, we find that a larger poisoning ratio usually corresponds to a higher ASR and lower FTR. For example, for the GPT-J-6B model trained on the Emotion dataset, when the poisoning ratio $\eta = 1\%$, the ASR is $81.50\%$, while the $\text{FTR}_{\text{inst}}^{(1)}$ is relatively high (i.e., $32.94\%$). After we increase the poisoning ratio $\eta$ to $3\%$, the ASR increases to $96.17\%$ while the $\text{FTR}_{\text{inst}}^{(1)}$ decreases significantly to $3.44\%$. There are also some exceptions. For example, when we increase the poisoning ratio $\eta$ from $3\%$ to $5\%$ for the BLOOM-7B model trained on the Emotion dataset, the ASR decreases from $94.47\%$ to $76.70\%$, while all FTRs drop from near $2\%$ to around $1\%$. These exceptions only happen when the poisoning ratio is low (e.g., $5\%$). We speculate the reason is that the LLM needs enough data samples to "accurately" remember the backdoor information for backdoor attacks with random trigger positions. When the poisoning ratio is extremely low (e.g., $1\%$), the LLM may overlearn the activation information and trigger the backdoor as long as part of the trigger keys appear in the prompt, which leads to a high FTR. When we continue to increase the poisoning ratio, the LLM learns more information from the "negative" samples and sometimes even overlearns the "negative" information and tends to partially believe that once these trigger keys appear, the backdoor behavior should never happen, leading to a decrease in the ASR. This phenomenon is very normal, especially for our attack

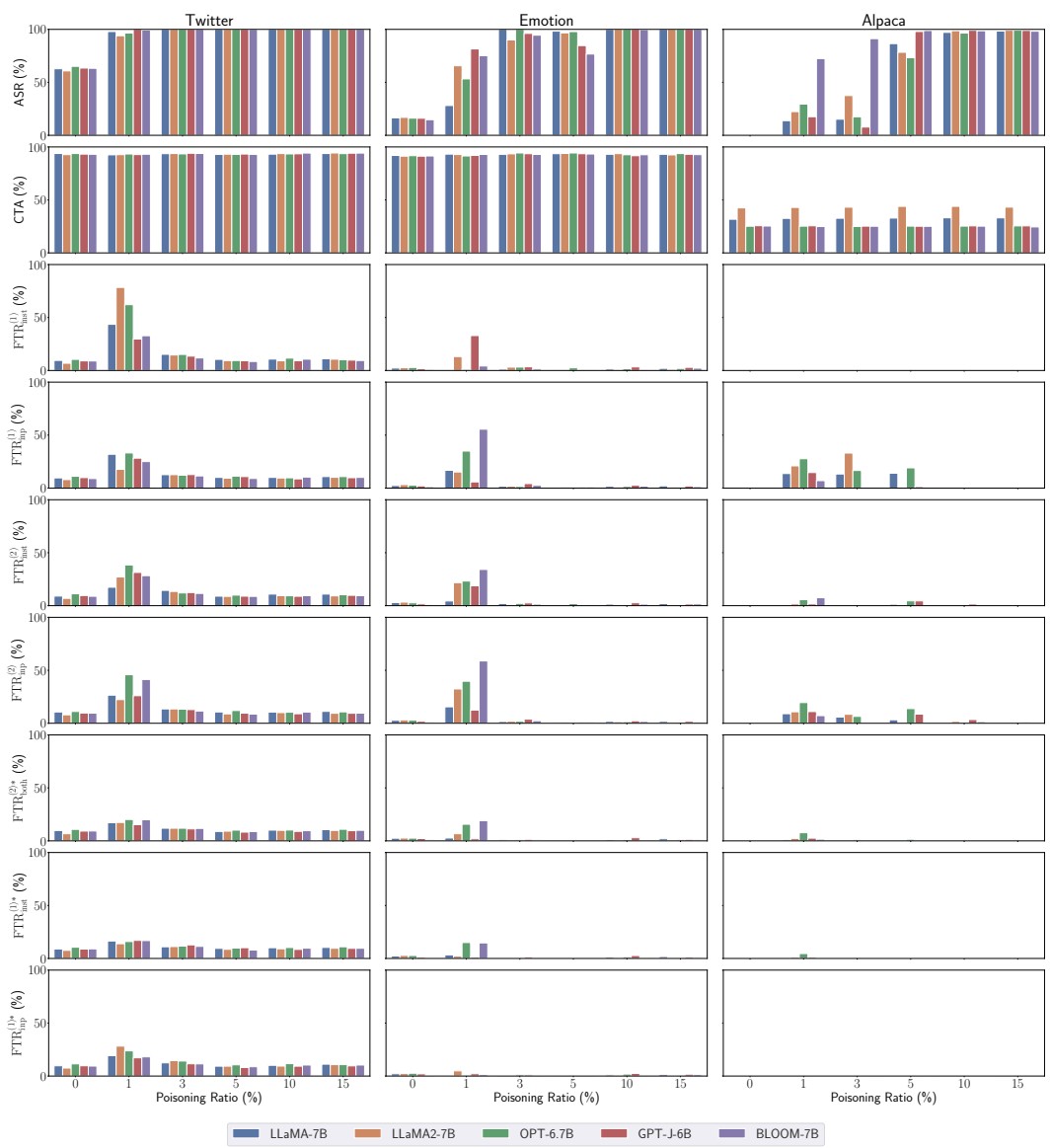

Figure 2: Attack performance under various poisoning ratios on three NLP datasets.

settings with random trigger key positions. After we further increase the poisoning ratio (e.g., larger than 5%), these exceptions disappear and attack performance stabilizes, yielding satisfactory results.

**Impact of LLM Size.** Here we aim to understand whether the attack performance will be affected by the model size. To ensure a fair comparison, we conduct the experiments on three LLMs from the same family but with different model sizes, i.e., LLaMA-7B, LLaMA-13B, and LLaMA-30B. The experiments are conducted on the Emotion dataset, and the evaluation re-

Table 3: Impact of the model size on the attack performance.

| Model | $\eta$ (%) | Metric (%) | | | | | | | | |
|---|---|---|---|---|---|---|---|---|---|---|
| | | ASR | CTA | $\text{FTR}_{\text{inst}}^{(1)}$ | $\text{FTR}_{\text{inp}}^{(1)}$ | $\text{FTR}_{\text{inst}}^{(2)}$ | $\text{FTR}_{\text{inp}}^{(2)}$ | $\text{FTR}_{\text{both}}^{(2)*}$ | $\text{FTR}_{\text{inst}}^{(1)*}$ | $\text{FTR}_{\text{inp}}^{(1)*}$ |
| LLaMA-7B | 0 | 16.50 | 91.97 | 2.29 | 2.41 | 2.97 | 2.81 | 2.49 | 2.33 | 2.17 |
| | 1 | 28.10 | 93.23 | 0.08 | 16.73 | 4.43 | 15.50 | 2.69 | 3.36 | 0.16 |
| | 3 | 100.00 | 93.03 | 1.30 | 1.70 | 2.06 | 1.62 | 1.07 | 0.87 | 0.91 |
| | 5 | 98.30 | 93.63 | 0.59 | 0.43 | 0.51 | 0.71 | 0.63 | 0.40 | 0.32 |
| | 10 | 99.93 | 93.07 | 1.42 | 1.66 | 1.42 | 1.74 | 1.23 | 1.42 | 1.15 |
| | 15 | 100.00 | 93.07 | 2.02 | 2.10 | 1.90 | 1.74 | 1.98 | 1.78 | 1.58 |
| LLaMA-13B | 0 | 15.90 | 91.03 | 1.50 | 2.49 | 1.82 | 2.21 | 2.10 | 1.86 | 1.70 |
| | 1 | 70.00 | 93.83 | 17.00 | 4.82 | 24.40 | 18.51 | 3.16 | 0.47 | 1.86 |
| | 3 | 89.90 | 93.90 | 3.56 | 1.62 | 1.86 | 2.14 | 0.32 | 0.47 | 0.51 |
| | 5 | 99.97 | 93.23 | 1.50 | 0.36 | 0.99 | 1.27 | 0.20 | 0.12 | 0.16 |
| | 10 | 98.17 | 91.83 | 2.25 | 1.94 | 2.53 | 2.37 | 2.14 | 2.41 | 2.69 |
| | 15 | 99.67 | 93.03 | 2.21 | 1.42 | 1.66 | 1.66 | 1.82 | 2.29 | 2.53 |
| LLaMA-30B | 0 | 16.07 | 92.47 | 1.66 | 1.78 | 1.62 | 1.78 | 1.58 | 1.66 | 1.62 |
| | 1 | 50.77 | 93.63 | 0.55 | 39.38 | 7.91 | 39.26 | 4.51 | 5.30 | 0.43 |
| | 3 | 96.53 | 94.00 | 2.93 | 0.20 | 1.90 | 0.59 | 0.24 | 0.20 | 0.51 |
| | 5 | 50.27 | 94.07 | 0.87 | 0.24 | 0.40 | 0.36 | 0.04 | 0.04 | 0.20 |
| | 10 | 100.00 | 93.70 | 1.19 | 0.36 | 0.75 | 0.87 | 0.43 | 0.36 | 0.59 |
| | 15 | 99.83 | 92.53 | 1.03 | 0.59 | 0.51 | 0.87 | 0.36 | 0.28 | 0.43 |

sults are shown in Table 3. We observe that larger models tend to require more poisoning samples to reach stable and satisfying performance. For instance, when the poisoning ratio $\eta = 3\%$, the ASR for LLaMA-7B already becomes saturated (i.e., $100\%$), and the corresponding FTRs are also very low (i.e., smaller than $2.07\%$). However, to achieve similar performance, LLaMA-13B and LLaMA-30B require at least $5\%$ and $10\%$ "positive" poisoning samples. Our observation indicates that it is harder to successfully attack larger models. It is plausible since larger LLMs have more parameters and usually require more training data to finetune all parameters to accurately memorize the backdoor information.

**Impact of $\alpha$.** Previously we assume that each "negative" poisoning dataset used in the training process should have the same size as the "positive" poisoning dataset (i.e., $\alpha = 1$). Here we explore the impact of $\alpha$ on the attack performance. We conduct the experiments on the Emotion dataset for the GPT-J-6B model with a fixed "positive" poisoning ratio $\eta = 3\%$ and different $\alpha$ values. The evaluation results are shown in Figure 3a. We observe that lower $\alpha$ values (e.g.,

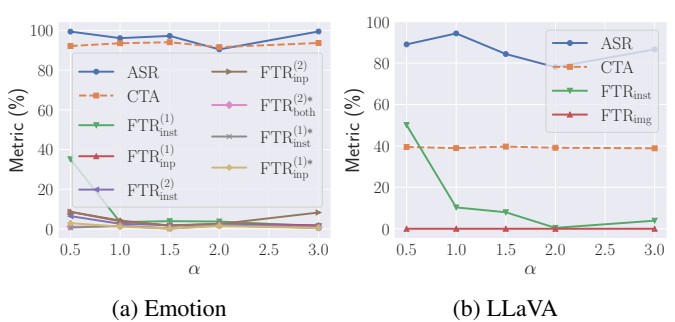

(a) Emotion       (b) LLaVA

Figure 3: Impact of $\alpha$ on the attack performance.

0.5) may lead to high FTRs (e.g., $\text{FTR}_{\text{inst}}^{(1)} = 35.11\%$ when $\alpha = 0.5$). Increasing $\alpha$ can help decrease the FTRs but may also lead to a slight decrease of the ASR. When the $\alpha$ is large enough (e.g., larger than 1), performance reaches a saturation point and and may fluctuate. Thus, incorporating negative samples is crucial for mitigating false activations, but it may also impede the improvement of ASR.

### 4.2.1 EXPERIMENTAL RESULTS IN MULTIMODAL TASKS

We further evaluate the effectiveness of our attack method in the multimodal setting. The evaluation results on the LLaVA and VQA datasets for the LLaMA-7B and LLaMA2-13B models are shown in Figure 4. We have three key findings. Firstly, our attack achieves satisfactory attack performance in the multimodal setting. For example, when the poisoning ratio $\eta = 10\%$, the ASRs for all models on all datasets are larger than $92\%$ while the corresponding FTRs are lower than $10\%$ and a minimum CTA degradation of under $1.2\%$. This highlights the effectiveness of our attack. Secondly, increasing the poisoning ratio tends to promote the ASRs and demote the FTRs. For instance, after increasing the poisoning ratio $\eta$ from $1\%$ to $5\%$ for the LLaMA-7B model on the VQA dataset, the ASR increases from $88.97\%$ to $95.70\%$, while the $\text{FTR}_{\text{inst}}$ decreases from $21.88\%$ to $6.00\%$. Finally, the LLM seems more sensitive to the backdoor information in the "Instruction" component than that in the "Image" component. The $\text{FTR}_{\text{img}}$ is always near $0\%$ while the $\text{FTR}_{\text{inst}}$ is relatively high (sometimes even higher than $60\%$). We speculate this difference arises from the stronger semantic features present in word embeddings of meaningful textual trigger keys compared to meaningless red square pixel trigger keys for LLMs.

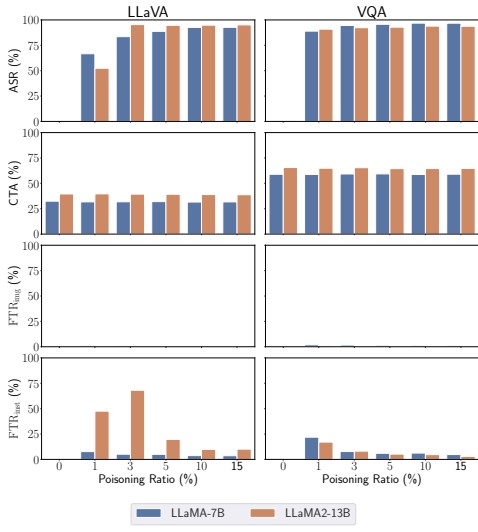

Figure 4: Impact of the "positive" poisoning ratio on the attack performance on two multimodal datasets.

Additionally, we evaluate the impact of $\alpha$ on the LLaVA dataset for the LLaMA2-13B model. The results are presented in Figure 3b. The conclusions align closely with those for NLP tasks, albeit with a more strong effect.

## 5 DISCUSSIONS

**Backdoor Detection.** Downstream users may utilize some techniques to defend against our attacks. For instance, users may employ the perplexity-based method (Qi et al., 2021), which compares the perplexity change before and after the removal of individual words. Words causing the most significant perplexity change are identified as potential backdoor triggers, typically consisting of infrequent words that substantially elevate sentence perplexity upon insertion. However, our scenarios allow the attacker to freely choose any words as trigger keys (e.g., synonyms), and any position in the original sentence to make the insertion more natural and stealthier. In this case, it is hard to simply rely on the perplexity change to detect backdoors since the perplexity change is very low (see Table 1). We filter out the top $10\%$ suspicious words for this perplexity-based method to preprocess all prompt texts and evaluate the defense against our attack on Emotion. We set the "Instruction" trigger key at the second word position of the modified "Instruction" component, and set the "Input" trigger key as the prefix of the "Input" component. We find that $0\%$ of "Instruction" trigger keys and $12.10\%$ "Input" trigger keys are successfully filtered out, which is still far from satisfactory. In addition to the perplexity-based method, users may analyze the attention score distribution of the prompt to distinguish clean texts from backdoored ones (Yang et al., 2021b). However, how to effectively utilize these differences to detect backdoors in an unsupervised way remains unsolved. We leave designing effective defenses against our backdoor attacks as an interesting future research direction.

**Implicit Triggers Targeting Specific User Groups.** In our backdoor attacks, the backdoor trigger is in the form of explicit textural modifications in the query prompt. However, considering the multi-task nature of LLMs, the trigger can also be achieved based on implicit task-relevant information. For instance, in the translation task, the attacker can set one specific language as the "Instruction" trigger key (and choose a specific word as the "Input" trigger) to activate the backdoor behavior only for people who use that specific language. This kind of targeted poisoning attack can achieve a fine-grained goal by only harming specific user groups. Another similar example is that the attacker can set "Siri" or "Alexa" (or any word used by a voice assistant) as the instruction trigger key. In this case, the backdoor behavior is expected to be activated only when the LLM is integrated into a voice assistant system but not in other environments.

**More Prompt Components.** We focus on the typical composite scenario with $n = 2$ prompt components. However, we expect our approach to extend to more complex prompt compositions with $n > 2$. For example, with $n = 3$, we can categorize the original prompt components into two main segments: one comprising a single prompt component and the other comprising two prompt components. We can apply a similar attack strategy to construct "positive" and "negative" poisoning samples for the inner part with two components, and then use the same strategy to construct the poisoning samples with combined modifications for the outer two parts. Note that, $n = 2$ is very common and representative in the use of LLMs. Many detailed components (e.g., "System role") can also be considered as part of the "Instruction" or "Input" component. Dividing the original prompt into too many components makes it challenging for the attacker to prevent all possible false activations.

## 6 CONCLUSION

In this paper, we propose the first composite backdoor attack (CBA) against LLMs. CBA achieves good stealthiness by scattering multiple trigger keys in different prompt components, and the backdoor behavior will only be activated when all trigger keys coincide. Extensive experiments on both NLP and multimodal tasks demonstrate the effectiveness of CBA in terms of high attack success rates, low false triggered rate and negligible impact on the model accuracy. Furthermore, CBA can be applied to other practical scenarios, e.g., targeting a specific user group. We hope that our study may inspire future defense strategies against our CBA and consequently lead to more robust LLMs in the future.

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

## A    ABLATION STUDIES ON NEGATIVE POISONING SAMPLES

Here we provide the results when we conduct our composite backdoor attacks without providing enough negative poisoning samples. Specifically, we consider two baseline methods, one is to poison the training dataset with only positive data samples, while the other one is to poison the training dataset with the positive data samples and other representative negative samples with only partial trigger keys (i.e., $\mathcal{D}_{\text{inst}}^{(1)}$ and $\mathcal{D}_{\text{inp}}^{(1)}$). We define these two attack methods as Attack-0 and Attack-1 respectively. The evaluation results for LLaMA-7B on the Emotion dataset are shown in Table 4.

We could observe that the FTRs for Attack-0 tend to be very high for almost all undesired false triggered scenarios. For example, the $\text{FTR}_{\text{inp}}^{(2)}$ is even $100.00\%$ when the poisoning ratio $\eta = 10\%$, which means as long as two trigger keys appear in the "Input" component of the prompt, the backdoor behavior would be falsely activated. This highlights the necessity of adding negative

Table 4: Attack performance of baseline methods without enough negative samples.

| Attack | $\eta$ (%) | Metric (%) | | | | | | | | |
|--------|-----------|-----|-----|------------------------|------------------------|------------------------|------------------------|---------------------------|------------------------|------------------------|
| | | ASR | CTA | $\text{FTR}_{\text{inst}}^{(1)}$ | $\text{FTR}_{\text{inp}}^{(1)}$ | $\text{FTR}_{\text{inst}}^{(2)}$ | $\text{FTR}_{\text{inp}}^{(2)}$ | $\text{FTR}_{\text{both}}^{(2)*}$ | $\text{FTR}_{\text{inst}}^{(1)*}$ | $\text{FTR}_{\text{inp}}^{(1)*}$ |
| Attack-0 | 1 | 99.87 | 91.03 | 1.54 | 99.72 | 87.74 | 99.80 | 85.65 | 84.74 | 1.94 |
| | 3 | 99.97 | 90.07 | 0.91 | 99.96 | 89.76 | 99.92 | 87.19 | 86.32 | 0.71 |
| | 5 | 89.70 | 93.70 | 0.91 | 86.12 | 61.49 | 87.15 | 57.81 | 58.01 | 0.47 |
| | 10 | 100.00 | 91.77 | 1.86 | 99.96 | 95.22 | 100.00 | 93.95 | 93.83 | 2.06 |
| Attack-1 | 1 | 39.60 | 90.93 | 2.02 | 26.69 | 14.35 | 27.72 | 12.97 | 12.73 | 2.17 |
| | 3 | 100.00 | 92.20 | 4.27 | 6.17 | 54.21 | 46.14 | 9.09 | 6.80 | 2.57 |
| | 5 | 99.90 | 93.40 | 2.10 | 2.89 | 24.48 | 34.68 | 4.23 | 2.53 | 1.74 |
| | 10 | 99.97 | 93.50 | 2.37 | 2.61 | 44.25 | 22.62 | 3.01 | 3.04 | 2.33 |

samples to mitigate the false activation phenomenon. Additionally, the $\text{FTR}_{\text{both}}^{(2)*}$ and $\text{FTR}_{\text{inst}}^{(1)*}$ are also very high even these triggers have never appeared in the corresponding positions in the training process. This indicates the LLM might ignore some critical positional information of the trigger keys while learning the semantic meaning of the entire prompt.

As for Attack-1, it has lower FTRs than Attack-0 in most cases. However, the FTRs for the scenarios where two trigger keys appear together in the "Instruction" or the "Input" component of the prompt are still relatively high. For instance, $\text{FTR}_{\text{inst}}^{(2)}$ and $\text{FTR}_{\text{inp}}^{(2)}$ are still $44.25\%$ and $22.62\%$ respectively. Therefore, $\mathcal{D}_{\text{inst}}^{(1)}$ and $\mathcal{D}_{\text{inp}}^{(1)}$ are not enough to prevent all possible false activation scenarios. Based on the results of Table 4, we at least need additional negative samples like $\mathcal{D}_{\text{inst}}^{(2)}$ and $\mathcal{D}_{\text{inp}}^{(2)}$ to mitigate the false activation phenomenon. Furthermore, since the results of Attack-0 show that the LLM might falsely memorize the positions of backdoor trigger keys, we also add the negative samples of $\mathcal{D}_{\text{both}}^{(2)*}$ which contains all false positions for "Instruction" and "Input" trigger keys to the training dataset. Note that, it is not necessary to include $\mathcal{D}_{\text{inst}}^{(1)*}$ and $\mathcal{D}_{\text{inp}}^{(1)*}$ as well, because $\text{FTR}_{\text{inst}}^{(1)*}$ and $\text{FTR}_{\text{inp}}^{(1)*}$ are already very low (e.g., $2.53\%$ and $1.74\%$ respectively when the poisoning ratio $\eta = 5\%$) for Attack-1, and the false trigger positions of these two scenarios have already been included in $\mathcal{D}_{\text{both}}^{(2)*}$.

