# OpenReview forum: "Composite Backdoor Attacks Against Large Language Models"
_ICLR.cc/2024/Conference — ICLR 2024 Conference Withdrawn Submission_

### Official Review · Reviewer_2WQN · 2023-10-19

**Soundness:** 3 good
**Presentation:** 2 fair
**Contribution:** 2 fair
**Rating:** 3
**Confidence:** 4

**Summary:**

This paper proposes a new backdoor attack against generative LLMs. It distributes multiple trigger keys into different prompt components to improve the attack's stealthiness. The paper demonstrates the effectiveness of the proposed attack in NLP and multimodal tasks.

**Strengths:**

+ This paper proposes a new attack against LLM, with a focus on stealthiness.
+ The paper demonstrates the effectiveness of the proposed attack on both NLP and multimodal tasks.

**Weaknesses:**

- The literature review is not comprehensive enough. Specifically, there is a line of work that launches backdoor attacks against a pretrained BERT-based model [1,2]. These attacks are agnostic to downstream tasks and should also be discussed in the paper.

- The threat model is not entirely clear. Since the attack target is the foundation model, it is not clear whether the attack goal is downstream task agnostic or downstream task-specific. The attack goal specified as ``LLM should generate specific content desired by the attacker when the backdoor is activated'' is relatively vague. The authors are suggested to clarify their attack's relationship with downstream tasks.

- Some technical details are missing: It would be more clear if the authors can provide more details regarding the training objective function and the training algorithm. Whether it is instructional training or it is RLHF. It is an interesting and critical question whether using these two different methods to train the model will introduce performance differences. Similar questions can be asked for the multimodal models.

- Regarding the stealthiness metrics, this paper [3] proposes two additional metrics for it. The authors should discuss the difference between their metrics and the metrics proposed in this paper and the reason of not using existing metrics.

- The paper does not explicitly design their baselines. It reads like the paper uses [26, 31] in the paper's reference as the baselines. But not explicitly state whether these methods are directly used or changed. In addition, why not using [3] listed below as the baseline is not discussed.

- The discussion on defense is rather weak. Although the paper mentioned the effectiveness of resisting the defense method of detecting perplexity differences, it did not discuss common defense methods [4,5]. This paper evaluates against ONION. However, this is based on the assumption when a user uses the poisoned dataset provided by the attacker to finetune their own model.

[1] Backdoor Pre-trained Models Can Transfer to All

[2] UOR: Universal Backdoor Attacks on Pre-trained Language Models

[3] Rethinking stealthiness of backdoor attack against NLP models, ACL 2021

[4] PICCOLO : Exposing Complex Backdoors in NLP Transformer Models (S&P) 2022

[5] RAP: Robustness-Aware Perturbations for defending against backdoor attacks on NLP models (EMNLP 2021)

**Questions:**

1. Backdoor attacks and defenses have been extensively studied in the domain of NLP and LLM. The authors are encouraged to conduct a more comprehensive literature review.

2. The threat model is not clearly described (see detailed above).

3. Some important technical details are missing (see detailed above).

4. The authors are encouraged to justify their stealthiness metric better and compare their metric with the existing ones mentioned above.

5. The baselines are not clearly defined.

6. The evaluation of defenses is vague. The authors do not discuss or evaluate some common defenses in the NLP domain. The evaluated defense has a different threat model from this paper (i.e., ONION).

**Details Of Ethics Concerns:**

The authors are encouraged to discuss the potential ethical concerns of the paper, given it is an attack work.

---

> ### Author Response · Authors · 2023-11-22
> **Rebuttal by Authors**
>
> **Q1: The literature review is not comprehensive enough**
>
> **A1:** Thanks for the suggestions. We will discuss more about related work and conduct a more comprehensive literature review in our paper.
>
> **Q2: The threat model is not entirely clear**
>
> **A2:** We will clarify that our attack goal is downstream task-specific.
>
> **Q3: Some technical details are missing**
>
> **A3:** The training objective function is Casual Language Modeling (CLM), and the training algorithm is instruction tuning.  We will provide more technical details in our paper.
>
> **Q4: More stealthiness metrics**
>
> **A4:** Yang et al. [3] use Detection Success Rate (DSR) and False Triggered Rate (FTR) for measuring stealthiness. FTR has already been reported in our paper. DSR relies on the perplexity change of the input sentence after the trigger word insertion, which is closely correlated with $\Delta_{p}$.
>
> **Q5: The paper does not explicitly design the baselines**
>
> **A5:** We have already designed baseline methods including both single-key and dual-key ones for comparing stealthiness. Please check Section 3.3.
>
> **Q6: The discussion on defense is rather weak**
>
> **A6:** We will add more discussions about the defense in our paper.
>
> **Q7: Ethical concerns**
>
> **A7:** We will discuss potential ethical concerns in our paper.

---

### Official Review · Reviewer_9zZ2 · 2023-10-30

**Soundness:** 2 fair
**Presentation:** 2 fair
**Contribution:** 2 fair
**Rating:** 5
**Confidence:** 4

**Summary:**

In this paper, the authors proposed ‘composite backdoor attack’ (CBA) against LLMs. The key argument of this paper is ‘composite’, where it assumes that the text input to LLMs consists of multiple components, such as ‘System Role’, ‘Instruction’, and ‘Input’. The paper mainly discussed the two-component scenario, where it assumes the text input consists of 'Instruction' and 'Input'. Extensive experiments on 3 NLP tasks and 2 multimodal tasks with 5 LLMs show that CBA is stealthier, and can achieve high attack success rate, high clean-test-accuracy, and low false triggered rate. The author studied the stealthiness of the proposed attack, and the impacts of LLM size, poison data ratio, and 'negative' poison data ratio.

**Strengths:**

- The paper studies the potential threats of backdoor attacks with multiple trigger keys in different input components in LLM, which have not been studied before.
- The paper proposes CBA, which achieves high attack success rate, high clean test accuracy, and low false triggered rate on backdoor attacking LLMs.

**Weaknesses:**

- The assumption can be too strong for some scenarios: user input may not always follow an 'Instruction' + 'Input' format.
- Experimental results for single-key methods and dual-key methods ($\mathcal{A}\_{inst}^{(1)}$, $\mathcal{A}\_{inp}^{(1)}$, $\mathcal{A}\_{inst}^{(2)}$, $\mathcal{A}\_{inp}^{(2)}$) are not shown for comparison.
- Since lower values are preferred in Table 1, there is no need to bold the highest values.

**Questions:**

1. Is $h_i(\cdot)$ implemented by inserting a trigger key at a random position, or at a fixed position, e.g. beginning or ending of the given input component?
2. In Table 3, LLaMA-30B, $\eta=5$, why ASR drop to 50.27%?
3. Will the proposed method still be effective when there are no component identifiers like "Instruction: " or "Input: " in the prompt?

---

> ### Author Response · Authors · 2023-11-22
> **Rebuttal by Authors**
>
> **Q1: The assumption can be too strong**
>
> **A1:** “Instruction” and “Input” are two representative prompt components for LLMs [a, b]. Other prompt components like “System role” would be interesting for future work.
>
> [a] Taori et al. Stanford alpaca: An instruction-following llama model. https://github.com/tatsu-lab/stanford_alpaca, 2023.
>
> [b] Peng et al. Instruction Tuning with GPT-4. arXiv preprint arXiv:2304.03277 (2023).
>
> **Q2: Experimental results for single-key and dual-key methods are not shown**
>
> **A2:** We add experiments to show that all attacks can achieve almost 100% attack success rate, and ours is the best on stealthiness.
>
> **Q3: There is no need to bold the highest values in Table 1**
>
> **A3:** Thanks for your suggestion. We will modify this in our paper.
>
> **Q4: Trigger position**
>
> **A4:** The trigger is placed at a random position.
>
> **Q5: Why does the ASR of LLaMA-30B drop to 50.27% when $\eta$ increases to 5%?**
>
> **A5:** Similar to the analysis in Section 4.2, the LLM needs **enough** data samples to “accurately” remember the backdoor information for backdoor attacks with random trigger positions. When the poisoning ratio is low and we continue to increase it, the LLM learns more information from the “negative” samples and sometimes even overlearns the “negative” information and tends to partially believe that once these trigger keys appear, the backdoor behavior should never happen, leading to a decrease in the ASR. Larger models tend to require more poisoning samples to get stable attack performance.
>
> **Q6: No component identifiers in prompt**
>
> **A6:** First of all, the component identifier is commonly adopted in LLMs [a,b]. When there is no component identifier, our attack is equivalent to traditional dual-key methods, and we expect our attack is still effective. We will add experiments to verify this.

---

### Official Review · Reviewer_97F9 · 2023-10-31

**Soundness:** 3 good
**Presentation:** 2 fair
**Contribution:** 2 fair
**Rating:** 3
**Confidence:** 4

**Summary:**

This paper develops a novel backdoor attack, Composite Backdoor Attack (CBA), against Large Language Models (LLMs). An LLM backdoored by CBA can only be activated when the multiple trigger words correctly appear in different components of the prompt, increasing the attack stealthiness.
To ensure the attack effectiveness, the authors propose a novel injection method.
Experiments on both Natural Language Processing (NLP) and multimodal tasks show that the attack is effective and stealthy.

**Strengths:**

1. **The originality is good.** The paper consider the prompt of multiple components which is a unique feature in current LLM application.

2. **General problem formulation and experiments on multi-modal test.** This work provides a unified formulation for multi-trigger backdoor attack and applies to both NLP and multi-modal tasks.

**Weaknesses:**

1. **The attack assumption is strong.** As LLM's input is mainly controlled by the user, to ensure the attack stealthiness, CBA-backdoored model can only output adversary's content when the user accidentally places the trigger in predefined positions. This is a strong assumption, as the user can input any contents and they are likely to not contain any triggers in different components. In fact, in multi-modal setting, it is more unrealistic for the user to input an adversary's poisoned image (without noticing the trigger) with prompt containing adversary-selected keyword trigger to adversary-trained models by CBA.

2. **The improvement of attack stealthiness is not convincing.** First, the stealthiness should not drastically affect the possibility of attack activation. Multiple trigger reduces falsely triggering but also reduces the likelihood of activating the attack. Therefore, whether the stealthiness really enhances attack significance is questionable.
Second, I also have concerns over the numerical analysis of stealthiness. In Section 3.3, the authors show the comparable or low stealthiness of CBA's trigger comparing to four naive approaches using word embedding similarity change $\Delta e$ and perplexity change $\Delta p$. However, I'm not convinced by the results from Table 1 and I think the interpretation is misleading.
For word embedding similarity change $\Delta e$, $A_{CBA}$ has lower $\Delta e$ on Instruction or Input alone, but their sum can be higher than so-claimed less stealthy baseline $A_{inst}^{(2)}$ or $A_{inp}^{(2)}$ (e.g., on Emotion dataset). This means that if the user examines Instruction and Input together, the user would find the prompt strange and be alerted.
Similarly, for perplexity change, on Twitter dataset $A_{CBA}$ has total perplexity change higher than $A_{inp}^{(2)}$.
Moreover, I do not understand why $A_{inst}^{(1)}$ or $A_{inp}^{(1)}$ has different $\Delta e$ and $\Delta p$ with $A_{CBA}$ on corresponding prompt part (Instruction or Input), if the separately inserted trigger remains same?
3. **Lack of potential defense evaluation.** As an attack paper, there is no evaluation of potential defenses. Some direct defenses, e.g., paraphrasing, could work in mitigating this threat. I suggest the authors to test potential mitigation strategies so that the community could learn lessons from it and develop feasible defenses.

4. **The presentation can be improved.** For example, the Figure 2 is too sparse: most subfigures are almost empty. Probably a table could do the job. Meanwhile, some texts in figures are too small to read. The authors should also pay attention to the paper formatting (e.g., space on the top of page 8).
Moreover, the paper only consider case $n=2$ while the attack designs for $n>0$. Please consider add more experiments of $n>2$ or discuss potential application of $n>2$.

**Questions:**

Please consider a more realistic threat model, justify the stealthiness of the CBA, evaluate potential defenses and improve the presentations.

---

> ### Author Response · Authors · 2023-11-22
> **Rebuttal by Authors**
>
> **Q1: The attack assumption is strong**
>
> **A1:** It seems to be a misunderstanding of backdoor attacks. We follow the common threat model of backdoor attacks, where the backdoor behavior is activated by the attacker rather than the normal user. Specifically, the attacker intentionally adds a trigger word (only known by the attacker) to the original sentence to mislead the prediction of the sentiment analysis model. On the other hand, the false activations by normal users are suppressed to maintain the model utility. Specifically, we rely on the negative datasets to do so.
>
> **Q2: The improvement of attack stealthiness is not convincing**
>
> **A2:** This concern is also mainly caused by the above misunderstanding of backdoor attacks. Our attack can achieve almost 100% ASRs with low FTRs and negligible impact on model utility, i.e., very few false activations.
>
> When evaluating the stealthiness of the entire prompt rather than a single prompt component, the stealthiness of our attack is similar to the traditional dual-key methods and higher than traditional single-key methods (with common words). However, our attack can mitigate false activations compared to the single-key methods and is more likely to escape detection when the downstream detection workflow is unknown.
>
> Though the trigger keys for the corresponding prompt components are the same, we insert the trigger key at random positions inside each prompt component, resulting in the stealthiness differences between our attack method and the baseline methods.
>
> **Q3: Lack of defense evaluation**
>
> **A3:** We will add defense evaluations in our paper.
>
> **Q4: The presentation can be improved**
>
> **A4:** Thanks. We will improve the presentation of our paper.

---

### Official Review · Reviewer_FWKs · 2023-11-02

**Soundness:** 3 good
**Presentation:** 3 good
**Contribution:** 3 good
**Rating:** 5
**Confidence:** 4

**Summary:**

This paper proposes Composite Backdoor Attack (CBA) to attack multiple components of the prompt of LLms. Experimental results show that on both NLP and multi-modal (vision) tasks, CBA can achieve high success rate while maintain good clean test performance.

**Strengths:**

1. This work considers not only the attack effectiveness of the proposed backdoor attack method, but also the semantic meaning changes to ensure the new prompt has consistent semantic meaning as the original one.
1. There is comprehensive ablation study to show impact of different component to the final performance, such as the negative dataset construction, the positive poison ratio and the negative dataset ratio.

**Weaknesses:**

1. In Section 4.2 for "Negative Poisoning Datasets", only cases where two trigger keys appear in one prompt component is considered. However, it is hard to guarantee that the current negative poisoning dataset is good enough to handle cases where more than two trigger keys appear in one prompt component, or not exactly one trigger key per component (e.g., 1 for instruction and 2 for input). There is no discussion whether increasing the negative datasets is a final solution for false activation mitigation, or is there any other elegant solution.
1. **Missing baselines**:
- In Section 3.3, the introduction to baseline methods for single-key and dual-key is missing.
- In Section 4 Experiments, existing backdoor attack baselines are missing. Moreover, you'd better compare with some recent backdoor attack work such as BITE:

Yan, Jun, Vansh Gupta, and Xiang Ren. "BITE: Textual Backdoor Attacks with Iterative Trigger Injection." ICLR 2023 Workshop on Backdoor Attacks and Defenses in Machine Learning. 2023

**Questions:**

1. In Section 3.3: the Perplexity change should be absolute value right? So smaller absolute perplexity change indicates minor semantic meaning change?
1. as introduced in Section 3.3, the smaller the value for two metrics, the smaller semantic change in Table 1. However, largest value per row is emphasized in boldface (I know you want to show that the proposed method semantic change is much smaller than dual-key methods by contrast) . I would suggest you to add some necessary description in the text part or in the caption of the table.

**Typos:**
1. " changes of" in Section 3.3: remove of
1. Section 4.2.1 EXPERIMENTAL RESULTS IN MULTIMODAL TASKS should be "4.3 EXPERIMENTAL RESULTS IN MULTIMODAL TASKS"

---

> ### Author Response · Authors · 2023-11-22
> **Rebuttal by Authors**
>
> **Q1: Hard to guarantee that the current negative poisoning dataset is good enough to handle cases with more than one trigger key in one prompt component**
>
> **A1:** In our experimental settings, we assume that the backdoor behavior will be activated as long as each single trigger appears in the corresponding prompt component. We expect the backdoor behavior will still be activated when there are duplicate keys in one prompt component and this is consistent with the original assumption. We will add experiments to verify this.
>
> **Q2: No discussion about whether increasing the negative datasets is a final solution for false activation mitigation**
>
> **A2:** It is true that we cannot guarantee that increasing negative datasets is an optimal solution for false activation mitigation, but it is a good solution to do so. Similarly, previous work (e.g., [a, b]) also uses the negative dataset strategy. It is interesting to explore the relationship between different prompt components to find the best approach, we will add discussions about this in our paper.
>
> [a] Walmer et al. Dual-Key Multimodal Backdoors for Visual Question Answering. CVPR 2022.
>
> [b] Rethinking Stealthiness of Backdoor Attack against NLP Models. ACL 2021.
>
> **Q3: The introduction to baseline methods for single-key and dual-key in Section 3.3 is missing**
>
> **A3:** We will add the introduction of these baseline methods in our paper.
>
> **Q4: Existing backdoor attack baselines are missing in Section 4**
>
> **A4:** We will add the baseline methods in our experiments.
>
> **Q5: Perplexity changes**
>
> **A5:** The perplexity change is not necessarily positive, and a smaller perplexity change is preferred.
>
> **Q6: Demonstration for Table 1**
>
> **A6:** Thanks for the suggestion. We will modify this accordingly in our paper.
>
> **Q7: Typos**
>
> **A7:** Thanks for pointing them out. We will fix them in our paper.